# High-Intensity Interval Training-Induced Hippocampal Molecular Changes Associated with Improvement in Anxiety-like Behavior but Not Cognitive Function in Rats with Type 2 Diabetes

**DOI:** 10.3390/brainsci12101280

**Published:** 2022-09-23

**Authors:** Amin Orumiyehei, Kayvan Khoramipour, Maryam Hossein Rezaei, Elham Madadizadeh, Manzumeh Shamsi Meymandi, Fatemeh Mohammadi, Mohsen Chamanara, Hamideh Bashiri, Katsuhiko Suzuki

**Affiliations:** 1Toxicology Research Center, Aja University of Medical Sciences, Tehran 1411718541, Iran; 2Neuroscience Research Center, Institute of Neuropharmacology, Department of Physiology and Pharmacology, Afzalipour School of Medicine, Kerman University of Medical Sciences, Kerman 7616914115, Iran; 3Department of Exercise Physiology, Faculty of Sport Sciences, Shahid Bahonar University, Kerman 7616913439, Iran; 4Sirjan School of Medical Sciences, Sirjan 7816916338, Iran; 5Department of Pharmacology, School of Medicine, Aja University of Medical Sciences, Tehran 1411718541, Iran; 6Physiology Research Center, Institute of Basic and Clinical Physiology Sciences, Department of Physiology and Pharmacology, Afzalipour Faculty of Medicine, Kerman University of Medical Sciences, Kerman 7616914115, Iran; 7Faculty of Sport Sciences, Institute of Sports Nutrition, Waseda University, Saitama 359-1192, Japan

**Keywords:** exercise, type 2 diabetes, anxiety-like behaviors, learning and memory, beta-amyloid, tau protein

## Abstract

(1) Background: Exercise exerts many neuroprotective effects in diabetes-induced brain disorders. In this study, we investigated the effect of high-intensity interval training (HIIT) on brain molecular changes and cognitive and anxiety-like behaviors in rats with type 2 diabetes. (2) Methods: Twenty-eight adult male rats were divided into four groups (*n* = 7): control (C), exercise + control (C+EX), diabetes (DM), and diabetes + exercise (DM+EX). Diabetes was induced using a two-month high-fat diet and a single dose of streptozotocin (35 mg/kg) in the DM and DM+EX groups. After, the C+EX and DM+EX groups performed HIIT for eight weeks (five sessions per week, running at 80–100% of V_Max_, 4–10 intervals) on a motorized treadmill. Then, the elevated plus maze (EPM) and open field test (OFT) were performed to evaluate anxiety-like behaviors. The Morris water maze (MWM) and shuttle box were used to assess cognitive function. The hippocampal levels of beta-amyloid and tau protein were also assessed using Western blot. (3) Results: The hippocampal levels of beta-amyloid and tau protein were increased in the DM group, but HIIT restored these changes. While diabetes led to a significant decrease in open arm time percentage (%OAT) and open arm enters percentage (%OAE) in the EPM, indicating anxiety-like behavior, HIIT restored them. In the OFT, grooming was decreased in diabetic rats, which was restored by HIIT. No significant difference between groups was seen in the latency time in the shuttle box or for learning and memory in the MWM. (4) Conclusions: HIIT-induced hippocampal molecular changes were associated with anxiety-like behavior improvement but not cognitive function in rats with type 2 diabetes.

## 1. Introduction

Diabetes mellitus is an impaired insulin secretion disorder with various degrees of peripheral insulin resistance, leading to hyperglycemia [1]. The most common form of diabetes mellitus is type 2, with a high prevalence (i.e., 90%) among adults. This prevalence is estimated to reach 1.5 million patients in 2030 [2], which will make type 2 diabetes (T2D) the seventh leading cause of death [3].

Diabetes causes significant complications such as nephropathy, vessel damage, and cardiovascular diseases [4,5]. Recent reports have demonstrated that hyperglycemia is closely related to the development of cognitive impairments and dementia [6]. The WHO reports that individuals with diabetes are six times more likely to develop dementia than healthy cases of the same age and gender [7,8]. According to previous studies, 75% of diabetic patients showed symptoms of learning and memory deficit as well as anxiety-like behaviors [7,8,9]. Alzheimer’s disease (AD), the leading cause of dementia, begins with impaired memory resulting from the accumulation of hyperphosphorylated tau protein and beta-amyloid (Aβ) [10,11]. Lopes et al. [12] showed that neuronal atrophy is tau-dependent in the stressed hippocampus. Hyperphosphorylated tau protein facilitates the detrimental effects of stress on brain structure and function, leading to anxiety-like behaviors and cognitive impairments.

In general, three methods have been proposed for the prevention and treatment of diabetes-induced AD-like symptoms, including diet therapy, medication, and exercise training [13,14]. Among them, exercise training has been emphasized because it does not have side effects, does not need special equipment, and is not expensive. Added to these are the psychological advantages associated with exercise training [13,14,15,16].

High-intensity interval training (HIIT) induces comparable chronic physiological adaptations with less expenditure of time than moderate-intensity continuous training (MICT), placing HIIT at the center of researchers’ attention in the last decade [17,18]. HIIT includes intervals (from 45 s to 4 min) of high-intensity activity (more than 80% of maximal heart rate) and low-intensity activity (approximately 50% of maximal heart rate) [19]. Kim et al. [20] showed that HIIT improves cognitive impairment in obese rats. It has been shown that physical activity can lower the levels of Aβ and hyperphosphorylated tau in the brain [21]. Endurance training is shown to inhibit the Aβ production pathway [22]. In addition, exercise has been reported to reduce Aβ deposition by adaptive regulation of amyloid precursor protein cleaved proteases, brain-blood transport proteins, degrading enzymes, and autophagy [23]. Dose-response studies have shown that higher intensity running may be more beneficial in reducing hyperphosphorylated tau accumulation in the brain [21,24]. Nonetheless, Elahi et al. [25] reported an increase in hyperphosphorylated tau accumulation following exercise.

Considering the increasing incidence of diabetes, which leads to cognitive and anxiety impairments, and the known benefits of HIIT as a cheap, available, and healthy treatment, we aimed to study the effect of eight weeks of HIIT on brain molecular changes and cognitive and anxiety-like behaviors in rats with T2D.

## 2. Materials and Methods

In this study, 28 male Wistar rats aged 8 weeks and weighing an average of 200 g were purchased from the animal farm of Kerman University of Medical Sciences. They were housed in special polycarbonate cages under controlled room temperature with an average of 23 ± 1 °C and humidity of 46–54% under a light–dark cycle of 12:12 h.

All animal handling and scarifying procedures were accepted by the ethics committee of AJA University of Medical Sciences (approval number: JAUMS.REC1400.132). The animals were randomly divided into four groups (N = 7 in each group): control (C), control + exercise (C+EX), diabetes (DM), and diabetes + exercise (DM+EX). Diabetes induced using two month high-fat-diet and a single dose of The rats in the DM and DM+EX groups were fed a high-fat diet for two months and a single dose of 35 mg/kg streptozotocin (STZ). Three days after injection of STZ, fasting blood glucose (FBG) were measured and the rats with FBG levels above 300 mg/dl were entered the study as diabetic rats [26]. we measured FBG the training period as well. Then, the rats in training groups underwent an eight-week exercise. Behavioral tests (i.e., open field test (OFT), elevated plus maze (EPM), Morris water maze (MWM), and shuttle box) were performed 24 h after the last training session. After these tests, twenty animals (four groups of five rats) were anesthetized, and hippocampal tissues were harvested to assess tau and Aβ protein levels using Western blot. The procedure timeline is presented in Figure 1.

### 2.1. High-Fat Diet

A high-fat diet was purchased from the Royan Research Institute in Isfahan, which included the following ingredients: 60% fat (245 g Lard and 25 g Soybean oil), 20% carbohydrate (125 g Lodex 10 and 72.8 g Sucrose), 20% protein (200 g Casein and 3 g Cysteine), 50 g fiber (SOLKA-FLOC), 50 g minerals, 3 g vitamins, and 0.5 g stain [27,28]. The composition of the regular diet was similar to that of the high-fat diet, except for the percent of fat and carbohydrate (Table 1) [28,29].

### 2.2. High-Intensity Interval Training Protocol (HIIT)

All animals were familiar with exercise to control the effect of familiarization period (running with 8 m/min,0% inclination for 5 days and 10 min/day). Then, the rats maximum speed (V_Max_) measured as previously describe in another study [30]. The rats’ V_Max_ was measured every two weeks, and the new V_Max_ was used to calculate the relative velocity for the next two weeks. The exercise protocol was designed in our lab and we named in K1 protocol (Table 2).

### 2.3. Open Field Test (OFT)

The open field apparatus is a transparent plexiglass structure with 90 × 90 × 30 cm dimensions. This device has two central and peripheral zones. Each rat, with the liberty to move around the test field, was placed in the central zone. The OFT was carried out to evaluate anxiety, motor function, and exploratory behavior in an unknown environment [31]. The following variables were measured during 5 min of the experiment: distance moved (cm), time passed in the central and peripheral zone (min), frequency (no), and the number of rearing and grooming. A video tracking system (Borj Sanat) was used.

### 2.4. Elevated Plus Maze (EPM)

The EPM test was performed to assess anxiety in animals. The routine method explained in previous studies was applied [32]. The EPM apparatus consisted of four woody arms, including two opposite open arms (50 × 10 cm) and two opposite closed arms (50 × 10 × 40 cm). The sides of the open arms were supplied with a 1-cm plexiglass margin to prevent the rats from falling. The device was fixed at 50 cm from the ground. The test started by putting the animal at the center square of the device. The animals’ movements were recorded by a video tracking system (Borj Sanat) for 5 min, while the proper light (100 W) was set at the height of 120 cm from the center of the apparatus. The main criteria computed in an EPM experiment include the time residing in each arm and the number of entrances, which is calculated as percentage of open arm entries (%OAE) and percentage of open arm times (%OAT).
%OAT = (OAT spent ÷ Total time in both arms) × 100 
%OAE = (OAE ÷ Total entries in both arms) × 100 

### 2.5. Morris Water Maze (MWM)

The Morris water maze is a reliable task to assess spatial learning and memory in rodents [33]. The maze is a large circular tank, 140 cm in diameter and 45 cm in depth, filled with water (22–24 °C). The rats ran away from water and onto a hidden platform (15 cm wide, 35 cm high), which was located 1.5 cm beneath the water level. The apparatus was surrounded by different visual cues on the wall of the room, and their place remained unaltered throughout the test period. The maze was separated into four quadrants, and the animals were randomly located in one of the four same quadrants. The animal’s behavior in the experiment was video recorded by a camera located above the center of the pool. The spatial learning and memory-related indices, such as the total time spent to detect the hidden platform (escape latency), were measured by a video tracking system software (Ethovision, Noldus Information Technology, Amsterdam, Netherlands). The training phase included three blocks with a 30-min interval, and each block comprised four successive trials. In each trial, rats were randomly dropped into the tank from a specified point of each quadrant. The rats were allowed to swim for 60 s to find the hidden platform. If the rat did not find the platform within 60 s, it was gently placed on it and left there for 10 s. The spatial learning was measured by the total time spent to detect the hidden platform (escape latency). During the learning phase, animals indoctrinate to find the hidden platform, and it has been displayed by the decrement in their swimming distance and escape latency across subsequent blocks of training. After the detection of the platform, the animal remained there for 20–30 s and then was caged for 20–30 s before the subsequent trial. The retention of spatial memory was assessed two hours after training trials by removing the platform in a 60-s probe trial. The total time spent in the target quadrant, which formerly contained the platform, was recorded as the indicator of spatial memory retention [34].

### 2.6. Shuttle Box Test

The shuttle box is a passive avoidance model. It is believed that the animal avoids trying a potentially painful circumstance when remembering a previous experience [35]. The shuttle box was made of two plexiglass boxes with the same dimensions (20 × 20 × 30 cm). The bright chamber was the safe one, and as soon as the animal went to the dark chamber, he encountered noxious electric shock stimuli. The experiment included three phases: habituation phase, acquisition trial, and long-term memory [7].

In the habituation phase, the animal was gently placed in the bright chamber. After five seconds, the barrier between the dark and bright chambers was pulled up, and the rat was free to move to the dark zone. The total time of a disposition was 100 s. When the animal entered the dark chamber, the obstacle fell into place immediately. Without any shock in this phase, we took the animal back to its cage after 10 s.

In the acquisition trial, we returned the animals to the bright chamber after thirty minutes. The difference between the acquisition trial and the habituation was the shock given to the rats in the dark chamber after the barrier was pulled up. A shock (50 Hz, 1 mA) was supplied for 3 s through the bottom of the dark chamber. Twenty seconds after the shock, we took the animal back to its cage. We repeated the same procedure at 2-min intervals again. This procedure was repeated several times until the animal avoided entering the dark chamber.

Twenty-four hours later, we tested the animals’ memory retrieval. We removed the obstacle twenty seconds after we placed them inside the bright chamber, and the step-through latency time (SLT) in seconds was recorded as a memory retrieval index. Animals with the highest level of memory usually needed 5 min for this part of the test. An entry was recorded when the four paws of the animal were inside a new zone in all our tests.

### 2.7. Western Blot

Forty-eight hours after the last training session, the animals were anesthetized by intraperitoneal injection of ketamine (80 mg/kg) and xylazine (10 mg/kg), and the hippocampal tissues were harvested. The hippocampus was washed in PBS solution. An ultrasonic homogenizer performed homogenization in RIPA buffer solution with protease inhibitor on ice. The homogenate was centrifuged at 4 °C at 13,000 rpm for 20 min, and the supernatant was kept at −80 °C. Then, Western blotting was used to measure the amount of Aβ and the phosphorylated form of the tau protein. The total protein concentration in the hippocampal samples was measured by the Lowry method, while bovine serum albumin was used as standard. After matching the concentrations, 40 μg of protein from each sample was mixed with a buffer sample. Then, electrophoresed was performed for 75 min using 11% SDS-PAGE gel. After that, the proteins separated in the gel were transferred to PVDF paper. The membrane was then incubated in a 2% block solution overnight (at 4 °C). In the next step, the membrane was quenched four times each, washed with TBST solution for 5 min, and incubated for 3 h with the primary antibody (SANTA CRUZ BIOTECHNOLOGY, INC, sc-28365 and sc-21796, Santa Cruz, California, USA) (concentration 1.200) for each of the mentioned proteins. Then, the membrane was exposed to a secondary antibody (SANTA CRUZ BIOTECHNOLOGY, INC, sc-2357 and sc-516102) (with a concentration of 1.1000) for 1 h. In the next step, immune detection was recorded using Chemi Doc XRS + imaging system (Bio-Rad Company, California, USA) and analyzed by ImageJ software [36]. β-actin was used as a control.

### 2.8. Statistical Analysis

The normality and homogeneity of variances were checked using the Shapiro–Wilk and Levene tests, respectively. A repeated measures ANOVA and Bonferroni post-hoc test was performed to compare the learning phase in the Morris water maze. All other data were analyzed using one-way ANOVA followed by an LSD post-hoc test. In this study, all data are reported as mean ± standard error. The *p* values less than 0.05 were considered the criterion for statistical significance. Graph pad prism version 8.00 (GraphPad Software, San Diego, CA, USA) was used for data analysis.

## 3. Results

### 3.1. The Fasting Blood Glucose Results

We assessed FBG to confirm our diabetes induction method. Our results showed that blood glucose was significantly increased after diabetes induction (2 months of high-fat diet and STZ injection) (month 2) compared with before at baseline (month 0) in the DM and DM+EX groups (*p* = 0.000), with no significant difference between these groups. In addition, the HIIT reduced blood glucose significantly (*p* = 0.000) (Figure 2).

### 3.2. The Effects of HIIT on the %Open Arm Time (%OAT( and %Open Arm Entry (%OAE) of Rats with T2D in the Elevated Plus Maze (EPM)

Figure 3 provides the results obtained from the %OAT and %OAE of the animals in the EPM as an assessment of anxiety-like behaviors. One-way ANOVA indicated that there was a significant difference among the groups of study for %OAT [F (3, 24) = 6.042, *p* = 0.0032] and %OAE [F (3, 24) = 4.818, *p* = 0.0092]. As shown in Figure 3A,B, the %OAT and %OAE of the diabetic rats were significantly decreased in comparison to the control group (*p* < 0.001 and *p* < 0.01, respectively), suggesting an anxiety-like effect for the T2D group (DM) in the EPM. Exercise could significantly increase these parameters in diabetic animals compared to the DM group (*p* < 0.05), indicating the anxiolytic effect of HIIT.

### 3.3. The Effects of HIIT on the Inner Zone Distance, Inner Zone Frequency, Inner Zone Time, and Rearing and Grooming of Rats with T2D in the Open Field Test (OFT)

Figure 4 provides the results obtained from the activity of the animals in the open field test. One-way ANOVA indicated that there was a significant difference among the groups of study for grooming [F (3, 24) = 4.26, *p* = 0.015]. No significant difference for inner zone distance [F (3, 24) = 1.783, *p* = 0.177], inner zone frequency [F (3, 24) = 2.380, *p* = 0.0947], inner zone time [F (3, 24) = 2.5, *p* = 0.076] and rearing [F (3, 24) = 1.424, *p* = 0.2604] among the groups was evident. As shown in Figure 4A,C, further analysis revealed that the grooming of the T2D group (DM) significantly declined (*p* < 0.05) in comparison to the control group (*p* < 0.01), which could indicate an anxiety-like behavior induced by the high-fat diet and STZ. Furthermore, HIIT significantly increased grooming in comparison to the DM group (*p* < 0.01), suggesting the effect of HIIT on reversing this parameter in diabetic animals.

### 3.4. The Effects of HIIT on the Learning of Rats with T2D in the Shuttle Box

Figure 5 provides the results obtained from the shuttle box test. The memory of the animals was assessed as passive avoidance by step-through latency time. One-way ANOVA indicated that there was not a significant difference among the groups of study for step-through latency [F (3, 24) = 2.357, *p* = 0.0970].

### 3.5. The Effects of HIIT on the Learning and Memory of Rats with T2D in the Morris Water Maze (MWM)

The effects of HIIT alone or in the presence of diabetes on spatial learning in the Morris water maze are presented in Figure 6. During the training phase, the animals found the hidden platform faster and at a shorter distance in the 3rd block than in the 1st block. During the learning phase, the animals are indoctrinated to find the hidden platform, as displayed by the decrement in their swimming distance and escape latency across subsequent blocks of training. A reduction in time represents progress in the acquisition of the learning task. The post-hoc analysis did not reveal a remarkable difference in the time to detect the platform among the groups in the first block (*p* > 0.05, Figure 6A), proposing no major defect of spatial learning for diabetic rats. As can be seen from Figure 6A in the 2nd and 3rd blocks, no significant differences in the time to find the platform among groups were detected. It should be noted that the statistics for the comparison among blocks have not been shown on the graph.

In the recall probe, memory performance was assessed by exploring the correct quadrant and studying the time in which the platform was situated during the training phase. One-way ANOVA revealed that there was not a dramatic difference among the groups for the time spent in the target quadrant [F (3, 24) = 1.459, *p* = 0.2506] (Figure 6B). The outcome, as measured by time spent in the target quadrant, indicated that the diabetes rats remembered the platform location similar to the control rats (*p* > 0.05, Figure 6B), suggesting no memory impairment in the diabetic rats. These measures, which show memory retention, were not significantly different between the control, control + exercise, and the diabetes groups (*p* > 0.05, Figure 6B).

### 3.6. The Effects of HIIT on the Tau and Aβ Protein Level in the Hippocampus of Rats with T2D

As demonstrated in Figure 7, tau and Aβ protein levels in the hippocampus showed a significant increase in the diabetic rats compared to the control group (*p* < 0.05), indicating the accumulation of these proteins. Furthermore, the levels of these proteins significantly decreased in the DM+EX group compared to the DM group (*p* < 0.05), showing the positive effect of HIIT in diabetic rats.

## 4. Discussion

The focus of the present study was to investigate the effect of eight-week HIIT on the hippocampal levels of tau and Aβ proteins and the cognitive and anxiety-like behaviors in rats with T2D. Our results demonstrated that induction of T2D using a high-fat diet and a single dose of STZ (35 mg/kg) caused anxiety-like behaviors observed in the EPM and OFT. The HIIT significantly improved parameters of anxiety-like behaviors in the EPM and restored normal grooming in the OFT, suggesting the induction of anxiolytic behaviors by HIIT in the diabetic rats. No significant difference was observed in the memory of different groups in the shuttle box and MWM. In addition, the changes in anxiety-like behavior were in line with tau and Aβ protein changes since the levels of tau and Aβ protein in the hippocampus of diabetic rats were reduced after HIIT.

In agreement with our findings, several studies have shown that patients with diabetes suffer from anxiety and depression symptoms twice as much as healthy cases of the same age and sex [37,38,39]. The high comorbidity between obesity and mental disorders, such as anxiety, can dramatically exacerbate metabolic and neurological symptoms [40]. A similar result has been seen in an animal study [41]. Aksu et al. [42] reported that diabetic rats had higher levels of anxiety and spent more time in the closed arms of the EPM and less time in the center part of the open field. However, in contrast to our findings, Hilakivi et al. [43] showed no anxiety and exploratory behavior in diabetic mice. The different species used in these studies could explain the inconsistent results. On the other hand, epidemiological sex differences in anxiety disorders have been well documented [44], thus, comparing the sex differences in response to HIIT should be considered in prospective studies.

HIIT improved anxiety in the DM+EX group since the %OAT and %OAE of diabetic rats in the EPM were significantly increased after exercise and became close to the control level (Figure 5). Consistent with our results, other studies [45,46] have reported that moderate-intensity exercise training for 5 and 10 weeks improved anxiety-like behavior in diabetic female Wistar albino rats and adult male Sprague-Dawley rats, respectively. A similar study showed that eight-week moderate-intensity running could reduce anxiety in middle-aged rats or increase it in adult rats [47]; thus, age should be considered when interpreting the effect of training on anxiety. Grooming was increased in diabetic rats with aerobic exercise (running on a treadmill at 70% of their maximal capacity for 45 min, five days per week for 12 weeks), meaning that exercise induced a significant anxiolytic effect [41].

Another part of the results showed that memory deficit was not observed in the diabetic rats, and HIIT could not induce a considerable difference among groups. Braszko et al. [48] suggested that the level of learning ability could modify the effect of environmental intervention on spatial learning and memory. In contrast to our data, others demonstrated that HIIT could attenuate cognitive, memory, and learning impairment in both animal [20,38,49,50,51] and human [34,51,52] subjects. What should be taken into account in interpreting these data is the engagement of the hippocampus in learning and memory [30].

Our study showed an increase in tau and Aβ protein levels in the hippocampus of diabetic rats. It is known that diabetes may lead to increased tau and Aβ protein levels in both humans and animals [53,54]. The HIIT decreased the levels of tau and Aβ protein significantly. In agreement, the amount of tau and Aβ protein in AD decreased in male rats running on a treadmill [55].

The interesting finding of this study was the association of improvement in anxiety-like behavior with the HIIT-induced reduction in tau and Aβ accumulation in diabetic rats, a relation that was not seen with memory behavior. Interestingly, endurance training inhibited anxiety-like behavior in the rat model of AD associated with increased Aβ deposition and tau hyperphosphorylation [55]. As mentioned earlier, we believe that the intensity of the exercise protocol could play a vital role.

Pentkowski et al. [56] reported that anxiety was increased without significant deficits in spatial memory. Additionally, in 1400 older people without AD, anxiety showed a positive correlation with Aβ deposition [57]. Further studies are warranted to clarify the exact mechanisms.

## 5. Conclusions

In summary, our data provide support for the beneficial effects of HIIT on improving anxiety-like behaviors associated with the high-fat diet and STZ dose. The HIIT-induced hippocampal tau and Aβ protein changes were associated with anxiety-like behavior improvement but not cognitive function in rats with T2D. Since we did not evaluate all possible signaling pathways, the precise mechanism of the effect of HIIT in an animal model of diabetes remains to be elucidated.

## Figures and Tables

**Figure 1 brainsci-12-01280-f001:**
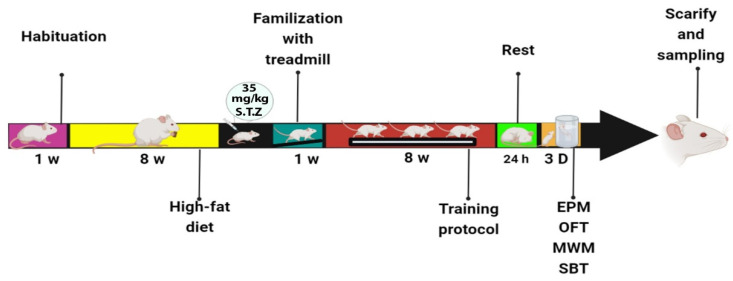
The procedure timeline of the study.

**Figure 2 brainsci-12-01280-f002:**
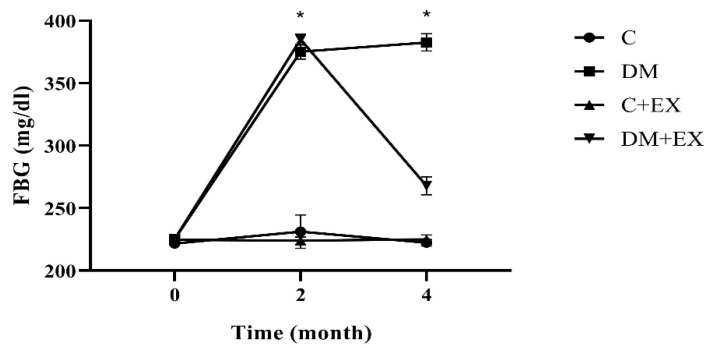
Fasting blood glucose before starting the intervention (month 0), after diabetes induction (2 months of high-fat diet and STZ injection) (month 2), and 48 h after the last training session (month 4) for all groups. Each bar represents mean ± SEM (*n* = 7 in each group). Groups: C, control; C+EX, control + exercise; DM, diabetes mellitus (type 2); DM+EX, diabetes mellitus (type 2) + exercise. FBG: Fasting blood glucose. * shows a significant difference between DM and DM+EX with other groups.

**Figure 3 brainsci-12-01280-f003:**
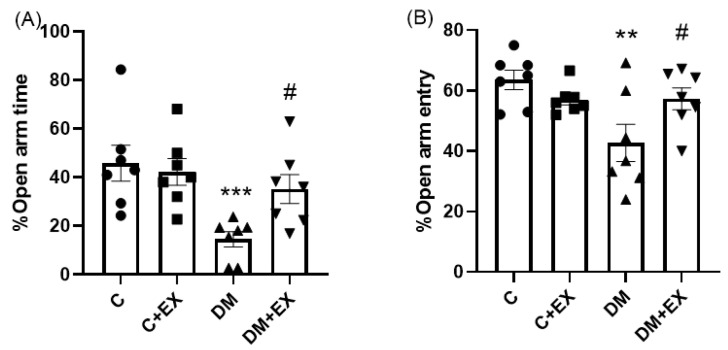
The effects of HIIT on the %open arm time (%OAT ((**A**), and %open arm entry (%OAE) (**B**) of rats exposed to high-fat diet and STZ in the elevated plus maze (EPM). Significant differences: ** *p* < 0.01, *** *p* < 0.001 as compared to the control group, ^#^
*p* < 0.05 as compared to the DM group. Each bar represents mean ± SEM (*n* = 7 in each group). Groups: C, control; C+EX, control + exercise; DM, diabetes mellitus (type 2); DM+EX, diabetes mellitus (type 2) + exercise.

**Figure 4 brainsci-12-01280-f004:**
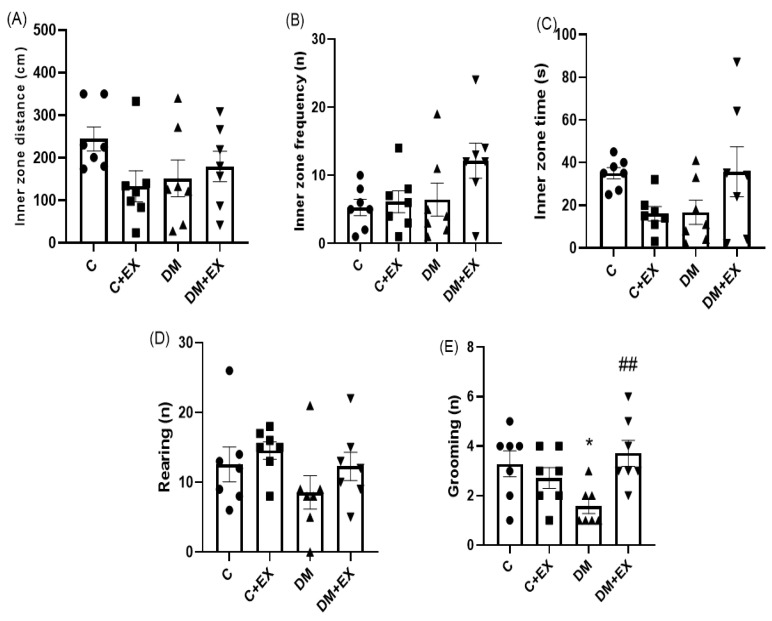
The effects of HIIT on the inner zone distance (**A**), inner zone frequency (**B**), inner zone time (**C**), rearing (**D**), and grooming (**E**) of rats exposed to high-fat diet and STZ in the open field test (OFT). Significant differences: * *p* < 0.05 as compared to the control group. ^##^
*p* < 0.01 compared to DM group. Each bar represents mean ± SEM (*n* = 7 in each group). Groups: C, control; C+EX, control + exercise; DM, diabetes mellitus (type 2); DM+EX, diabetes mellitus (type 2) + exercise.

**Figure 5 brainsci-12-01280-f005:**
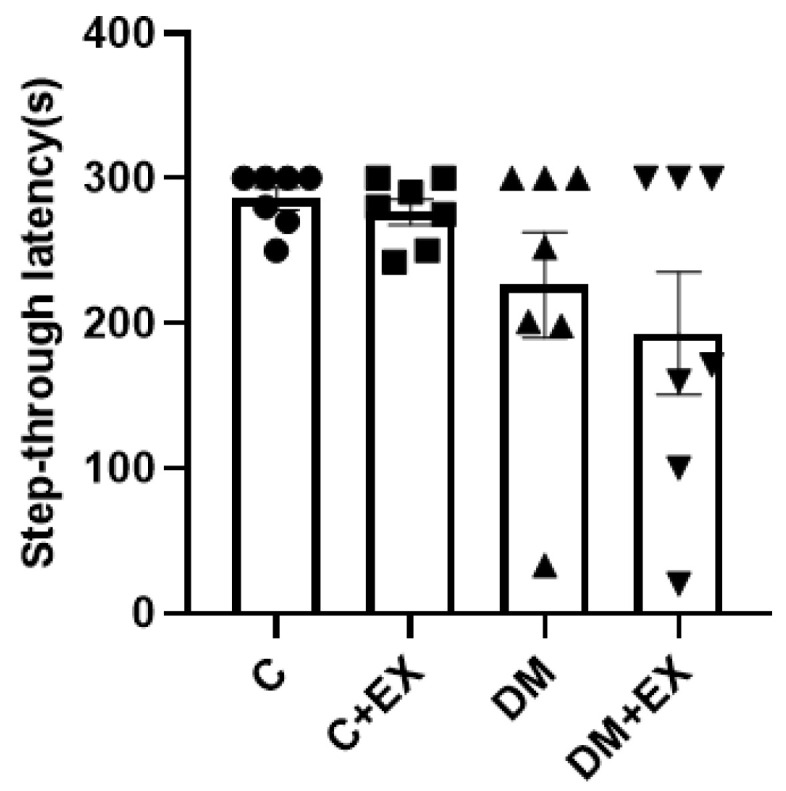
The effects of HIIT on step-through latency time in shuttle box test of rats exposed to high-fat diet and STZ. Each bar represents mean ± SEM (*n* = 7 in each group). Groups: C, control; C+EX, control + exercise; DM, diabetes mellitus (type 2); DM+EX, diabetes mellitus (type 2) + exercise.

**Figure 6 brainsci-12-01280-f006:**
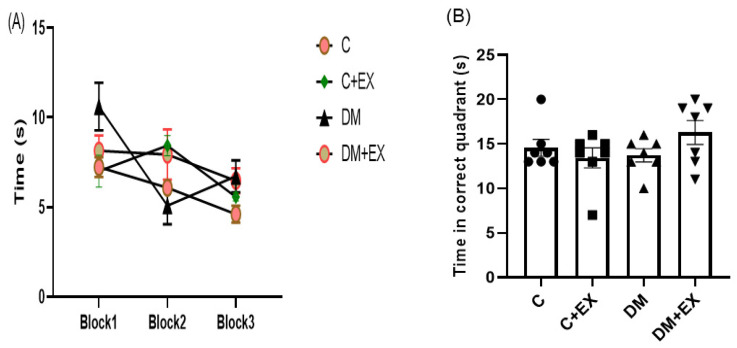
The effects of HIIT on time spent in each block (learning) (**A**) and time spent in the target quadrant (memory) (**B**) of rats exposed to high-fat diet and STZ in the Morris water maze (MWM). Each bar represents mean ± SEM (*n* = 7 in each group). Groups: C, control; C+EX, control + exercise; DM, diabetes mellitus (type 2); DM+EX, diabetes mellitus (type 2) + exercise.

**Figure 7 brainsci-12-01280-f007:**
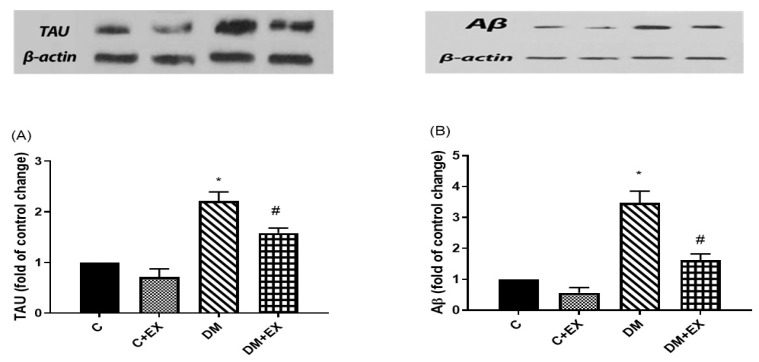
The effects of HIIT on tau (**A**) and Aβ (**B**) protein levels in the hippocampus of rats exposed to high-fat diet and STZ in the passive avoidance test (PAT). * *p* < 0.05 as compared to the control group, ^#^ *p* < 0.05 as compared to the DM group. Each bar represents mean ± SEM (*n* = 5 in each group). Groups: C, control; C+EX, control + exercise; DM, diabetes mellitus (type 2); DM+EX, diabetes mellitus (type 2) + exercise.

**Table 1 brainsci-12-01280-t001:** Regular and high-fat diet formulations.

Diet Ingredients	Fat	Carbohydrate	Protein	Fiber	Mineral	Vitamin
Regular diet	10%	70%	20%	50 g	50 g	3 g
High-fat diet	60%	20%	20%	50 g	50 g	3 g

**Table 2 brainsci-12-01280-t002:** HIIT protocol.

Week	Intervals	High-Intensity Interval Duration (min)	Low-Intensity Interval Duration (min)	High-Intensity Interval Velocity (%V_Max_)	Low-Intensity Interval Velocity (%V_Max_)	Total Exercise Time in a Session (min)
1	4	2	1	80	50	12
2	4	2	1	85	50	12
3	6	2	1	85	50	18
4	6	2	1	90	50	21
5	8	2	1	90	50	24
6	8	2	1	95	50	24
7	10	2	1	95	50	30
8	10	2	1	100	50	30

Slope was 0 and frequency was 5 in all weeks.

## Data Availability

https://kmu.ac.ir/en, accessed on 30 January 2022.

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
