# Peer review of "High-Intensity Interval Training-Induced Hippocampal Molecular Changes Associated with Improvement in Anxiety-like Behavior but Not Cognitive Function in Rats with Type 2 Diabetes"

_brainsci, 2022, doi:10.3390/brainsci12101280_

Round 1

Reviewer 1 Report

Here Amin Orumiyehei and colleagues investigated the effect of high-intensity interval training (HIIT) on brain molecular changes, cognitive and anxiety-like behaviors in rats with type 2 diabetes. And they concluded that there is a correlation by applying several molecular, pharmacological, and behavioral tests.

This work is informative that worth publishing within Metabolites readership and community. However, there are some minor concerns to be addressed before acceptance for publication.

First page left section:

Citation: Lastname, F.; Lastname, F.; Lastname, F. Title. Int. J. Environ.

Res. Public Health 2022, 19, x. https://doi.org/10.3390/xxxxx

It seems like there is a typo regarding the Journal name, it should be Brain Sciences, not Int. J. Environ. Res. Public Health.

Materials: How the authors choose the male rats as studying subjects, why not the females. It is highly suggested that the authors disclose or provide a reference (Bangasser, D. A., & Cuarenta, A. (2021). Sex differences in anxiety and depression: circuits and mechanisms. Nature Reviews Neuroscience, 22(11), 674-684.).

Regarding the experimental design, it is critical to discuss that high fat diet of 8 weeks might induce anxiety and depression in rat and this phenotype might affect the following treatment and training diagram. It is recommended to cite the following reference for these issues and the behavioral OFT EPM paradigms: Xia, G., Han, Y., Meng, F., He, Y., Srisai, D., Farias, M., ... & Wu, Q. (2021). Reciprocal control of obesity and anxiety–depressive disorder via a GABA and serotonin neural circuit. Molecular psychiatry, 26(7), 2837-2853.

Also, it seems like the result of control rat in the behavioral study is not quite match up with previous publications. For example, the open arm time (40%) and open arm entry (60%) is not consistent. The vital priority in behavioral tests is the control group. The authors should reanalyze the data and double check the literature for these baseline parameters.

Collectively, Minor Revision is recommended for this fabulous job!

Author Response

Please find attached our reply to the respected reviewer's comments. 

Reviewer 2 Report

The manuscript studies the effect of HIIT on anxiety-like behavior, learning, memory, and hippocampal Tau (A) and Aβ(B) protein levels using rats. The authors conducted behavioral tests and western blot to address the problem. Their results indicate an anxiety-like behavior in diabetes-induced rats, which can be reversed by HIIT. They also found changes of protein levels in hippocampus.

Major concerns:

1.       The author proposed that rats from C+EX and DM+EX groups went through 8-weeks HIIT exercise. How to guarantee the rats do not fall from the treadmill? It is amazing that they can stay on the track and run the whole session without any falling or escape. From table 2, It seems that the low intensity interval duration (min) was at least 80 min. The number of intervals was at least 4. Does that mean the rats can run continuously for 5 hours? That seems to be unlikely. How to control them?

2.       As mentioned above, the HIIT Protocol, i.e. table 2 is very unclear.  Why Low intensity interval velocity (V max) is higher than High intensity interval velocity (V Max)? Why do Intervals and High intensity interval duration not add up to Total exercise time in a session? Please remake the table and describe more in the method part.

3.       The number of animals used for the experiments are confusing. The author indicates 48 male Wistar rats were purchased. However, they also mentioned that the animals were randomly divided into four groups (N=7 in each group), which means 28 rats were used eventually. Where are the additional 20 rats? In addition, at line 101, it says twenty animals (four group of five rats) were anesthetized. Why not all of them?

4.       The conclusion about learning and memory is not consistent and not correct. Line 310, “…declined time and distance to find a submerged platform in the first block…”, not the case. Line 333, “Other part of results showed that spatial learning, but not spatial memory deficit happens in diabetic rats…” From Figure 4 and Figure 5, it is also not the case. There is no learning deficit in DM. Line 334, “HIIT improved learning and memory in DM+EX group.” Where is the data? Line 359, “T2D memory and anxiety like behavior (but not learning) impairment was developed.” Totally wrong. Please double check Figure 5. If there is a significance, please mark it. If not, please re-draw the conclusion.

5.       There is an “Original Images for Blots and Gels Requirements” for the journal. Please provide the original blots.

6.       Simple bar-plot is not acceptable. Please provide individual data points in the figures.

Minor concerns:

1.       Please double check the Figure numbers in the main context. Line 225, “As shown in Figure 1 (A, B) …”, should be Figure 2; Line 243, “As shown in Figure 2A and 2C …”, should be Figure 3.

2.       “After 24 hours of the last training session behavioral tests (i.e., Open Field Test (OFT), Elevated Plus Maze (EPM), Morris Water Maze (MWM) 100 and Shuttle box were performed.” From Figure 1, rest period seems to be 48h instead of 24h.

3.       For Western blot, please add the information about company and catalog number for all antibodies used.

4.       For MWM experiment, please indicate what to do if rats did not find the platform within 60s.

5.       There is a discrepancy from Figure 2 and Figure 3. From Figure 2, the anxiety level of C+EX is comparable with C, but from figure 3, the anxiety level of C+EX is increased compared to C. Please interpret and discuss.

6.       As mentioned above, how to interpret the difference observed between C+EX and C groups in Figure 3? There was no difference between DM+EX and DM though. Exercise seems to have different influence here. Please interpret and discuss.

Author Response

Please find attached our reply to respected reviewers' comments. 

Reviewer 3 Report

The manuscript by Orumiyehei et al. investigated whether HIIT could improve cognitive functions and ameliorate anxiety like behavior in diabetic rats, using behavioral and molecular assays. They found that diabetic rats has elevated Beta-amyloid and Tau levels in the hippocampus, which can be reduced by HIIT; and they found that diabetic rats has elevated anxiety levels, which can be improved by HIIT as well. The experiments are carefully designed and performed, the majority of data are well presented. However, there is major concerns over authors’ data interpretation and conclusions.

1.     Authors concluded: “In summary, in T2D memory and anxiety like behavior (but not learning) impairment was developed. Also, the level of Tau protein and Aβ were increased. But HIIT could restore all of these changes”. This statement is not fully supported by the data. There is no data supporting that memory is affected in T2D, and no data supporting that HIIT could restore.

2.     The results of OFT and EPM are both established measurement of anxiety-like behaviors in rodents. However, the results of these two test does not match. Especially that in OFT, HIIT and DM alone both induce more anxiety-like behavior in normal animals, making it hard to interpret whether DM+HIIT helps reduce anxiety in DM rats. The authors need to discuss further how they interpret the data and explain the discrepancy in two tests.

3.     In the abstract, the authors wrote “HIIT-induced hippocampal molecular changes were associated with anxiety-like behavior behavior improvement, but not cognitive function in rats with type 2 diabetes”. This statement does not match their conclusions in the end. They need to reconcile their findings and conclusions.

4.     Authors used HFD+streptozocin to induce T2D in rats, however, they did not show blood glucose level, GTT results to confirm that the rat they are working with actually develop T2D. Please provide.

Minor points:

1.     The authors need to better explain their experimental design for MWM and shuttle box text. Especially if they want to distinguish learning and memory retrieval capability in their study objects, they need to explain which parameter they are looking for that indicate these ability.

2.   Please proof read the manuscript and correct grammatical errors.

Author Response

Please find attached our reply to reviewers' comments. 

Round 2

Reviewer 2 Report

 Although I like the idea of the study, I cannot trust the results. Their conclusion was not consistent, they made obvious mistakes in description and their data was even not correct. 

It is hard for me to trust the data. They even changed the control group during revision. If possible, I’d like to have them to show the training video (how rats run on the treadmill). We previously did the similar thing with mice, which is very difficult to achieve. I’m not sure if the animals can keep up with the training protocol. If not, how to guarantee they reach to the required heart beats. Thank you very much!

Reviewer 3 Report

The revised manuscript has addressed all of my previous concerns, I am good with this paper being published.